# Reactive oxygen species oxidize STING and suppress interferon production

**Lili Tao[1], Andrew Lemoff[2], Guoxun Wang[1], Christina Zarek[1], Alexandria Lowe[1], Nan Yan[1,3], Tiffany A Reese[1,3]***

[1]Department of Immunology, University of Texas Southwestern Medical Center, Dallas, United States; [2]Department of Biochemistry, University of Texas Southwestern Medical Center, Dallas, United States; [3]Department of Microbiology, University of Texas Southwestern Medical Center, Dallas, United States

**Abstract** Reactive oxygen species (ROS) are by-products of cellular respiration that can promote oxidative stress and damage cellular proteins and lipids. One canonical role of ROS is to defend the cell against invading bacterial and viral pathogens. Curiously, some viruses, including herpesviruses, thrive despite the induction of ROS, suggesting that ROS are beneficial for the virus. However, the underlying mechanisms remain unclear. Here, we found that ROS impaired interferon response during murine herpesvirus infection and that the inhibition occurred downstream of cytoplasmic DNA sensing. We further demonstrated that ROS suppressed the type I interferon response by oxidizing Cysteine 147 on murine stimulator of interferon genes (STING), an ER-associated protein that mediates interferon response after cytoplasmic DNA sensing. This inhibited STING polymerization and activation of downstream signaling events. These data indicate that redox regulation of Cysteine 147 of mouse STING, which is equivalent to Cysteine 148 of human STING, controls interferon production. Together, our findings reveal that ROS orchestrates anti-viral immune responses, which can be exploited by viruses to evade cellular defenses.

## Introduction

Reactive oxygen species (ROS) are reactive chemicals generated primarily in mitochondria as a byproduct of oxidative metabolism (*Schieber and Chandel, 2014*). Due to their capacity to inactivate DNA, proteins and lipids, ROS induce cell death and defend cells against many pathogens. However, accumulating evidence suggests that ROS also control cellular signaling pathways. ROS regulation of signal transduction allows cellular pathways to rapidly adapt to changes in the oxidative environment.

Even though some pathogens are effectively controlled by ROS, other pathogens thrive in a cellular environment where ROS are abundant (*Paiva and Bozza, 2014*). For instance, DNA viruses such as Kaposi's sarcoma associated herpesvirus (KSHV), herpes simplex virus-1 (HSV-1) and Epstein Barr virus (EBV), all induce oxidative stress in cells. Moreover, treatment with antioxidants such as N-acetyl-cysteine (NAC) reduces viral burden (*Paiva and Bozza, 2014*). In the case of herpesviruses, ROS not only enhance replication, but also induce virus reactivation from latency and potentially contribute to virally induced cancers (*Bottero et al., 2013*; *Chen et al., 2018*; *Gao et al., 2018*; *Gonzalez-Dosal et al., 2011*; *Ma et al., 2013*; *Ye et al., 2011*). The underlying mechanism for ROS promotion of virus replication and reactivation remains to be investigated. One possibility is that ROS regulate the signaling pathways that activate the immune response to these viruses, in particular, the DNA sensing pathways.

DNA normally localizes to the nucleus, and the presence of DNA in the cytosol serves as a universal danger signal to activate pattern recognition receptors (PRRs) that distinguish self from non-self. Upon virus infection, cytosolic DNA is recognized by DNA sensors such as cyclic GMP-AMP synthase

***For correspondence:**
tiffany.reese@utsouthwestern.edu

**Competing interests:** The authors declare that no competing interests exist.

(cGAS), which catalyzes formation of an atypical cyclic di-nucleotide second messenger 2′,3′-cGAMP. 2′,3′-cGAMP binds and activates the Stimulator of Interferon Genes (STING) to induce production of type I interferon (IFN) and stimulate an immune response that promotes virus clearance (*Motwani et al., 2019*; *Wu and Chen, 2014*). Structural analysis of STING suggests that STING polymerization is necessary for its activation and that some cysteine residues may mediate STING polymerization by forming intermolecular disulfide bonds (*Ergun et al., 2019*; *Jin et al., 2010*; *Shang et al., 2019*). The chemical nature of cysteines is such that these residues are regulated by redox modifications, such as oxidation. However, post-translational modifications on STING cysteines have not been identified.

Because herpesviruses are DNA viruses that induce ROS and engage the cGAS/STING pathway, we hypothesized that ROS antagonize the production of interferon downstream of cGAS/STING during herpesvirus infection. Here, we found that ROS increased replication of murine gammaherpesvirus-68 (MHV68), a close genetic relative of KSHV and EBV. We also found that ROS suppressed interferon production in a STING-dependent manner. We further showed that the murine STING cysteine residue, C147 (equivalent to human C148), was oxidized upon ROS-inducing menadione treatment and that this cysteine was required for oxidation-sensitive inhibition of STING. Collectively, our results suggested that redox modification of STING is an important regulatory mechanism for STING activity during viral infection.

## Results

### ROS promote herpesvirus replication in macrophages

To determine if ROS promote a cellular environment conducive to virus replication, we used menadione as a tool to manipulate the level of ROS in cells. Menadione, also known as vitamin K3, is partially reduced by complex I in the mitochondria. The resulting semiquinone then participates in a redox cycle to partially reduce molecular oxygen, which generates ROS (*Iyanagi and Yamazaki, 1970*). Although prolonged treatment of a high dose of menadione leads to cell death, we determined a dose of menadione in bone marrow-derived macrophages (BMDMs) that induced no significant cell death (*Figure 1A*). Further, short-term treatment with menadione did not induce significant cell death even at a relatively high dose (*Figure 1A*). Menadione treatment is reported to reduce the glutathione/glutathione disulfide (GSH/GSSG) ratio and increase accumulated cellular ROS (*Chuang et al., 2002*; *Loor et al., 2010*). Consistent with these reports, we observed decreased gene expression of glutathione-disulfide reductase (*Gsr*) and glutamate-cysteine ligase regulatory subunit (*Gclm1*) with low-dose of menadione treatment. Therefore, menadione treatment of macrophages resulted in elevated oxidant levels in the cells (*Figure 1B*).

Because many viruses exploit ROS to facilitate their replication, we tested if increased ROS in macrophages affects growth of MHV68. To test the effects of ROS on virus replication, we first treated macrophages with low doses of menadione for sixteen hours, which increased cellular ROS while maintaining cell viability before virus infection. Macrophages were then infected with MHV68 at a multiplicity of infection (MOI) of 5 and virus growth was determined at indicated time points. Menadione treatment increased replication of MHV68 in a dose-dependent manner (*Figure 1C*). Hydrogen peroxide ($H_2O_2$) is a membrane permeable ROS, which induces secondary ROS production in cells upon extended treatment (*Fisher, 2009*). We pretreated macrophages with $H_2O_2$ at different concentrations for sixteen hours. The pretreatment was done in media containing FBS to maintain a cellular environment conducive to virus replication. We then infected macrophages with MHV68 at MOI of 5. Twenty-four hours after infection, cells expressing MHV68 lytic proteins were quantified using flow cytometry (*Reese et al., 2014*). Treatment with $H_2O_2$ increased the percentage of lytic protein positive cells in a dose-dependent manner (*Figure 1D*). Therefore, ROS induced by oxidants promoted MHV68 replication in macrophages.

### ROS inhibit interferon response upon STING activation

We next determined whether ROS antagonized the antiviral response to promote herpesvirus replication. Because of the central role of interferons in controlling viral replication, we asked if ROS promoted MHV68 replication by inhibiting the interferon response. We first tested whether menadione treatment impacts virus growth in wildtype (WT) control macrophages and type I interferon receptor

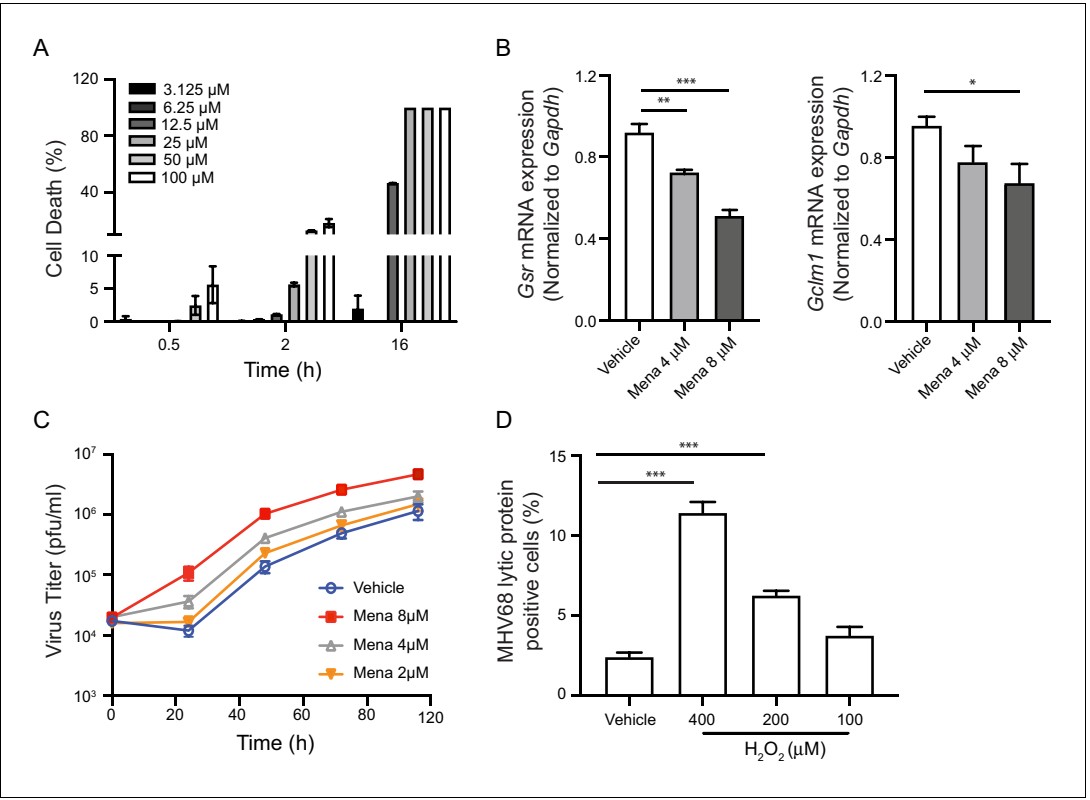

**Figure 1.** ROS promote herpesvirus replication in macrophages. (**A**) Fully differentiated bone marrow-derived macrophages (BMDMs) were treated with vehicle or different concentrations of menadione as indicated. Cell viability was determined at 0.5 hr, 2 hr and 16 hr after treatment. n = 2 with two technical repeats each time. (**B**) BMDMs were treated with vehicle, 4 μM or 8 μM menadione for 16 hr. Transcripts of *Gsr* and *Gclm1* were determined using qRT-PCR. n = 6. (**C**) BMDMs were treated with vehicle or different concentrations of menadione (mena) for 16 hr and then infected with MHV68 at multiplicity of infection (MOI) = 5. Virus titer was determined by plaque assay at 0 hr, 24 hr, 48 hr, 72 hr and 96 hr after infection. n = 3 with three technical repeats each time. (**D**) BMDMs were treated with vehicle or different concentrations of $H_2O_2$ for 16 hr in culture medium containing 10% fetal bovine serum, then infected with MHV68 at MOI = 5. Twenty-four hours after infection, cells were fixed and cells expressing virus lytic proteins were determined by flow cytometry. n = 3 with two technical repeats each time. Data are shown as mean ± SE, an ordinary one-way ANOVA was performed followed by Dunnett's multiple comparison test, only the p value for the most relevant comparisons are shown for simplicity. *, $p<0.05$, **, $p<0.01$, ***, $p<0.001$.

knockout (*Ifnar1*[-/-]) macrophages (*Muller et al., 1994*). While menadione treatment robustly increased virus growth in WT macrophages, it did not increase virus growth in *Ifnar1*[-/-] macrophages, suggesting that menadione interfered with the interferon response during virus infection (*Figure 2A*). Consistent with this idea, macrophages treated with menadione had significantly fewer transcripts of *Ifnb* and interferon stimulated genes (ISGs), *Cxcl10, Ccl5, Isg20* and *Isg15* (*Figure 2B and C*).

Because interferon responses are induced by multiple PRR signaling pathways after virus infection, we tested which PRR pathways were inhibited by ROS. Menadione treatment inhibited *Ifnb* expression upon interferon stimulatory DNA (ISD) stimulation, which engages the cGAS pathway. In contrast, menadione did not significantly alter the interferon response induced by poly I:C (sensed primarily by RIG-I) or poly dA:dT (sensed by multiple PRRs) (*Figure 2—figure supplement 1*). These results suggested that menadione-induced ROS selectively inhibited the cGAS-STING induced interferon response.

Sensing of cytosolic DNA involves the well-characterized cGAS signaling axis, which involves downstream components such as STING, TBK1 and IRF3. To further pinpoint which step of the

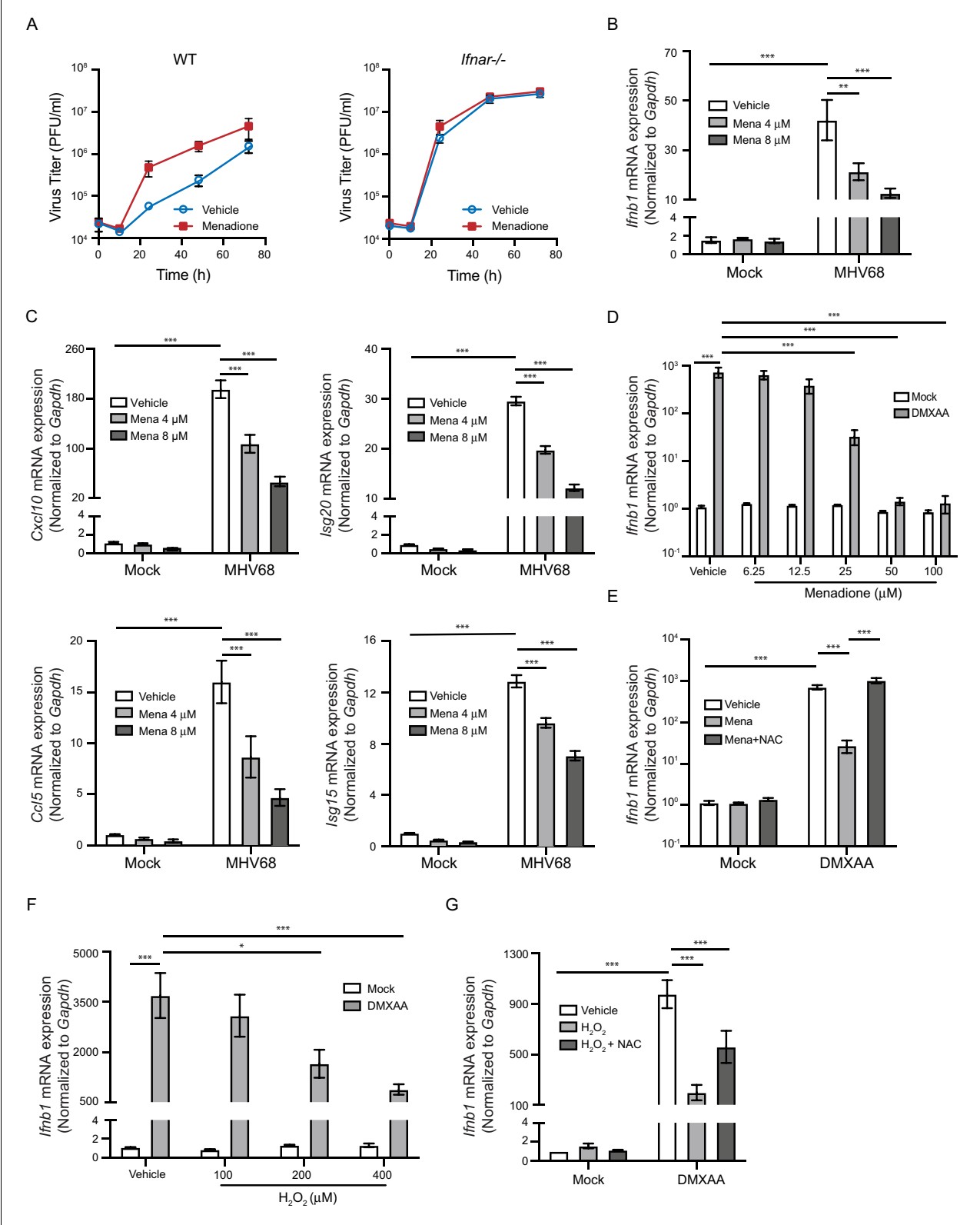

**Figure 2.** ROS inhibit interferon response upon STING activation. (**A**) BMDMs isolated from WT or *Ifnar-/-* mice were treated with vehicle or 8 μM menadione for 16 hr, then infected with MHV68 at MOI = 5. Virus titer was determined by plaque assay at 0 hr, 10 hr, 24 hr, 48 hr and 72 hr after infection. n = 1 with three technical repeats. (**B, C**) BMDMs were treated with vehicle, 4 μM or 8 μM menadione for 16 hr, then infected with MHV68 at MOI = 5. Transcripts of *Ifnb* (**B**) or ISGs (*Cxcl10, Isg20, Ccl5, Isg15*) (**C**) were determined at 6 hr after infection. n = 6. (**D**) BMDMs were treated with

*Figure 2 continued on next page*

*Figure 2 continued*

vehicle or different concentrations of menadione as indicated for 30 mins, then stimulated with DMXAA at 1 µg/ml. Transcripts of *Ifnb* were determined at 2 hr after stimulation. n = 3. (E) BMDMs were treated with vehicle, 25 µM menadione or 25 µM menadione and 2 mM NAC for 30 mins, then stimulated with 1 µg/ml DMXAA. Transcripts of *Ifnb* were determined at 2 hr after stimulation. n = 3. (F) BMDMs were treated with vehicle or different concentrations of $H_2O_2$ in serum free medium for 10 mins, then stimulated with 1 µg/ml DMXAA. Transcripts of *Ifnb* were determined 2 hr after stimulation. n = 3. (G) BMDMs were treated with vehicle, 200 µM $H_2O_2$ for 10 mins or 200 µM $H_2O_2$ for 10 mins followed by 5 mM NAC for 30 mins, then stimulated with 1 µg/ml DMXAA. Transcripts of *Ifnb* were determined 2 hr after stimulation. n = 4. Data are shown as mean ± SE, statistical analysis was conducted using two-way ANOVA followed by Tukey's multiple comparison test, only the p value for the most relevant comparisons are shown for simplicity. *, $p < 0.05$, **, $p < 0.01$, ***, $p < 0.001$.

The online version of this article includes the following figure supplement(s) for figure 2:

**Figure supplement 1.** ROS regulate interferon response in macrophages upon STING activation.
**Figure supplement 2.** ROS regulate interferon response in macrophages upon STING activation with 2′,3′-cGAMP.
**Figure supplement 3.** ROS do not suppress interferon in primary fibroblasts.

cGAS-STING pathway is regulated by ROS, we directly induced STING activation with DMXAA and 2′,3′-cGAMP, two murine STING ligands. *Ifnb* expression induced by DMXAA and 2′,3′-cGAMP was strongly inhibited by different doses of menadione (*Figure 2D*, *Figure 2—figure supplement 2*). Furthermore, treatment with the antioxidant N-acetyl-L-cysteine (NAC) restored *Ifnb* expression when combined with menadione (*Figure 2E*). To directly induce ROS in cells, we pulsed macrophages with $H_2O_2$ for 10 min prior to DMXAA stimulation. Similar to menadione treatment, $H_2O_2$ dose-dependently repressed transcription of *Ifnb* after DMXAA stimulation (*Figure 2F*). Addition of NAC after H2O2 partially rescued *Ifnb* expression (*Figure 2G*). Collectively, these data suggest that ROS directly inhibited interferon production downstream of cGAS sensing.

Our next question was whether ROS inhibit interferon production in non-macrophage cell types. ROS production is a critical effector mechanism for macrophages to fend off microbial challenges (*Van Acker and Coenye, 2017*; *Fang, 2011*). However, the high levels of ROS produced by macrophages requires the existence of intrinsic protective mechanisms against ROS, which otherwise would result in premature death of these immune cells during inflammatory responses. Indeed, macrophages are equipped with multiple mechanisms that allow them to be more resistant to ROS than other cell types (*Virág et al., 2019*). To test whether ROS inhibition of interferon production is a general immunoregulatory mechanism that functions in cells other than macrophages, we pulsed primary fibroblasts with $H_2O_2$ at the same concentration as we used in macrophages, followed by activation of STING with DMXAA. The expression of *Ifnb* in fibroblasts was not repressed, but rather slightly increased by $H_2O_2$ treatment (*Figure 2—figure supplement 3*). This is consistent with the notion that MEFs are more susceptible to ROS, which induces mitochondrial DNA fragmentation and primes interferon response (*West et al., 2015*). Therefore, ROS may negatively regulate interferon response in a cell-type and concentration dependent manner.

## Endogenous ROS regulate interferon response

Because ROS are constantly generated by cellular respiration, cells contain low levels of ROS in the absence of exogenous stimulation. We therefore tested if endogenous ROS regulate the interferon response after STING activation. We first pretreated mouse macrophages with NAC to deplete endogenous ROS, then infected cells with MHV68. We observed elevated *Ifnb* and ISG expression with NAC treatment (*Figure 3A and B*). We also observed increased *Ifnb* transcripts when macrophages were cotreated with NAC and DMXAA, compared with DMXAA alone (*Figure 3C*). Peroxisomes are metabolically active organelles and are an important source of ROS in macrophages. ACOX1 is the rate-limiting enzyme that metabolizes long chain fatty acid in peroxisomes and is a major producer of $H_2O_2$. Using macrophages from *Acox1*$^{-/-}$ mice (*Fan et al., 1996*), we observed increased *Ifnb* expression upon DMXAA stimulation compared to those isolated from the WT littermate controls (*Figure 3D*). We did not observe an inhibition in MHV68 virus growth by NAC pretreatment or in *Acox1*$^{-/-}$ macrophages compared with wildtype (*Figure 3—figure supplement 1*). We speculate that the differences we observed in interferon levels caused by neutralizing basal ROS may not be sufficient to confer a viral growth difference, but these differences could be more

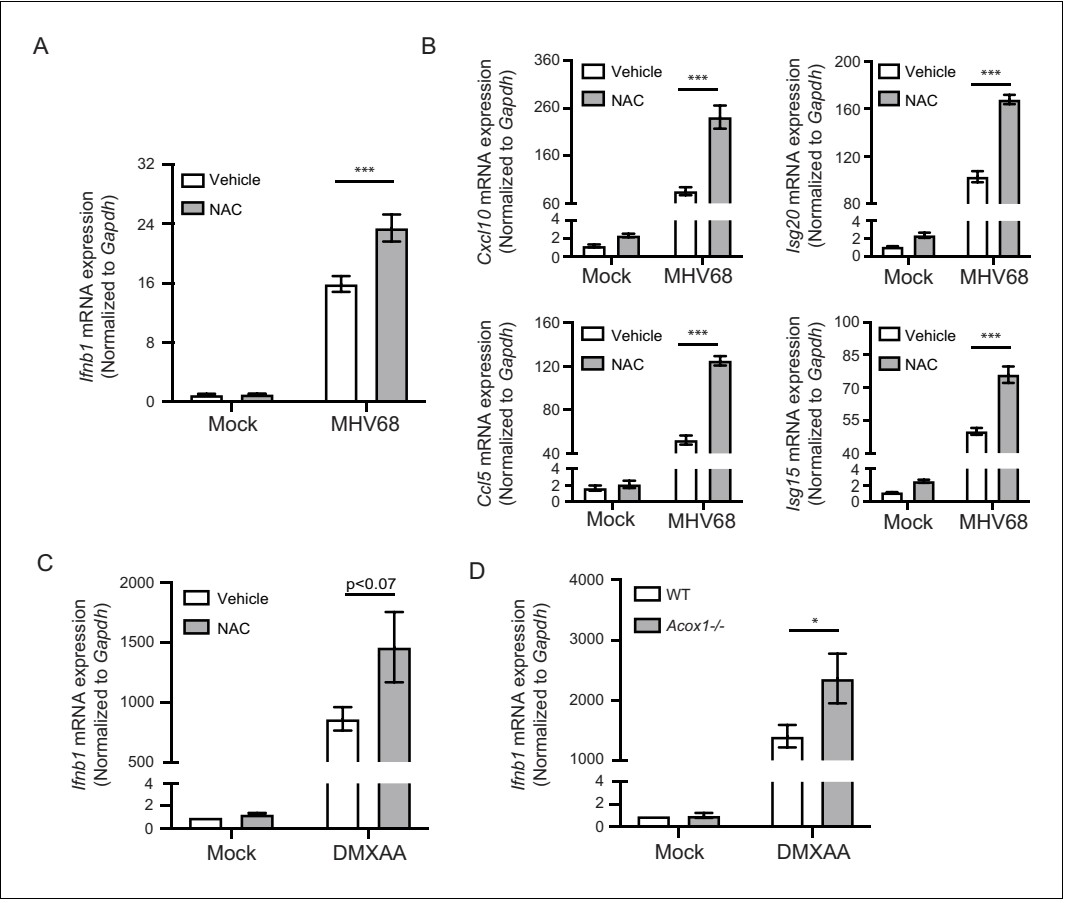

**Figure 3.** Endogenous ROS regulate interferon response upon STING activation. (**A, B**) BMDMs were treated with 2 mM NAC for 30 min, then infected with MHV68 at MOI = 5. Transcripts of *Ifnb* (**A**) or ISGs (*Cxcl10, Isg20, Ccl5, Isg15*) (**B**) were determined 6 hr after infection. n = 6. (**C**) BMDMs were treated with 2 mM NAC for 30 min, then stimulated with 1 μg/ml DMXAA. Transcripts of *Ifnb* were determined 2 hr after stimulation. n = 4. (**D**) BMDMs isolated from *Acox1-/-* or WT littermate control were stimulated with 1 μg/ml DMXAA. Transcripts of *Ifnb* were determined 2 hr after stimulation. n = 4. Data are shown as mean ± SE, statistical analysis was conducted using two-way ANOVA followed by Tukey's multiple comparison test, only the p value for the most relevant comparisons are shown for simplicity. *, $p < 0.05$, **, $p < 0.01$, ***, $p < 0.001$.

The online version of this article includes the following figure supplement(s) for figure 3:

**Figure supplement 1.** Inhibition of endogenous ROS has no effect on virus replication.

significant in the context of a genetic background that predisposes to elevated interferon and autoimmunity.

## ROS regulate interferon response by inhibiting STING polymerization

We next determined how ROS regulate STING-induced interferon production. STING is a transmembrane protein anchored on the endoplasmic reticulum (ER) as a dimer in the absence of stimulation. Upon activation, STING undergoes a conformational change and rearranges to form a polymer (*Ergun et al., 2019*; *Ishikawa et al., 2009*; *Tanaka and Chen, 2012*). It is then transported to the Golgi complex where it recruits TBK1, leading to TBK1 phosphorylation. STING also serves as a scaffolding protein to specify phosphorylation of IRF3 by phosphorylated TBK1 (pTBK1). Phosphorylated IRF3 (pIRF3) translocates to the nucleus and induces IFNβ transcription (*Wu and Chen, 2014*). To analyze activation of this pathway, we induced STING activation with DMXAA after treating BMDMs with $H_2O_2$ for 10 mins, and measured protein expression of STING, TBK1, and IRF3 at 0 mins, 30

mins, 60 mins and 90 mins after stimulation (*Figure 4A*). H₂O₂ treatment did not change the basal protein levels of STING, TBK1 or IRF3. However, pTBK1 and pIRF3 were significantly inhibited by H₂O₂ treatment. We also found that H₂O₂ inhibited TBK1 phosphorylation in a dose-dependent manner (*Figure 4B*). When using menadione to induce cellular ROS, pTBK1 and pIRF3 were also inhibited by menadione treatment (*Figure 4C*). These data suggest ROS inhibit interferon signaling upstream of TBK1 activation, possibly by inhibiting STING activation.

Next, we measured herpesvirus growth to test whether STING is required for the inhibitory effect of ROS on interferon. To this end, we quantified virus growth in WT and STING-deficient macrophages (*Sauer et al., 2011*). Menadione increased virus growth in WT macrophages but had no effect on virus growth in STING-deficient macrophages, supporting our hypothesis that ROS inhibited interferon in a STING-dependent manner (*Figure 4D*).

We subsequently determined if polymerization of STING was regulated by ROS, because STING polymerization is an early event in STING activation that occurs prior to translocation to the ER. In the absence of stimulus, STING monomers spontaneously form dimers. This process does not involve covalent linkage between the monomers. In contrast, STING polymerization requires the formation

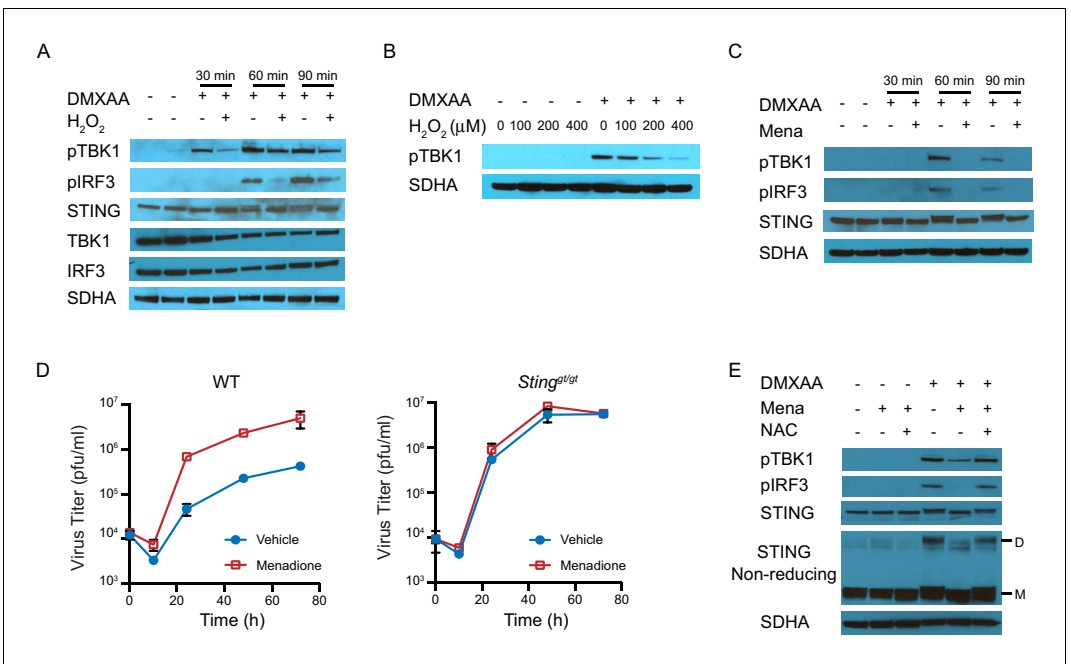

**Figure 4.** ROS regulate interferon response by inhibiting STING dimerization. (**A**) BMDMs were treated with vehicle or 200 µM H₂O₂ for 10 min in serum free culture medium, then stimulated with 1 µg/ml DMXAA. Western blots of TBK1, IRF3, STING, pTBK1 and pIRF3 were performed at 0 min, 30 min, 60 min and 90 min after stimulation. Data shown are representative of 2 independent experiments. (**B**) BMDMs were treated with vehicle or different concentrations of H₂O₂ in serum free culture medium for 10 mins, then stimulated with 1 µg/ml DMXAA. Level of pTBK1 was determined at 60 min after stimulation. n = 1 (**C**) BMDMs were treated with vehicle or 25 µM menadione for 30 min, then stimulated with 1 µg/ml DMXAA. Western blot of TBK1, IRF3, STING, pTBK1 and pIRF3 was performed at 0 min, 30 min, 60 min and 90 min after stimulation. Data shown are representative results of two independent experiments. (**D**) BMDMs isolated from WT control or *Sting^{gt/gt}* mice were treated with vehicle or 8 µM menadione for 16 hr, then infected with MHV68 at MOI = 5. Virus titer was determined by plaque assay at 0 hr, 10 hr, 24 hr, 48 hr and 72 hr after infection. n = 3 with three technical repeats each time. (**E**) BMDMs were treated with vehicle, 25 µM menadione, 25 µM menadione and 2 mM NAC for 30 min, then stimulated with 1 µg/ml DMXAA. STING polymerization was determined by non-reducing SDS-PAGE. M: STING monomer; D: STING dimer. Data shown are representative of 2 experiments.

The online version of this article includes the following figure supplement(s) for figure 4:

**Figure supplement 1.** Schematic diagrams of STING monomer, dimer and polymer on different electrophoresis gels.

**Figure supplement 2.** ROS decrease TBK1 recruitment during STING activation.

of interdimer disulfide bonds (*Ergun et al., 2019*). Therefore, polymerized STING appears as a dimer on non-reducing SDS-PAGE, as SDS disrupts only the non-covalent interactions, leaving covalent disulfide bonds between the monomers intact (*Figure 4—figure supplement 1*). As expected, DMXAA induced STING polymerization (shown as a dimer) and phosphorylation of TBK1 and IRF3. However, polymerization was significantly inhibited by menadione treatment, and restored by the addition of NAC (*Figure 4E*). As a result of decreased STING polymerization, the recruitment of TBK1 upon DMXAA stimulation was also diminished (*Figure 4—figure supplement 2*). In conclusion, ROS regulate interferon signaling by inhibiting STING polymerization.

## ROS oxidization of STING at Cysteine-148 blocks STING activation

The next question we addressed was whether ROS oxidized STING, thus inhibiting polymerization. Recent structural studies suggest that STING polymerization and activation require formation of an intermolecular disulfide bond at Cysteine 148 (*Ergun et al., 2019*). The ability of C148-C148 disulfide bonds to bridge STING dimers suggests that the residue harbors a free thiol prior to stimulation. Because free thiol functional groups are susceptible to oxidation, we hypothesized that ROS inhibit STING function by oxidizing this free thiol group. To test this hypothesis, we treated human STING (hSTING) overexpressing fibroblast cells with menadione or hydrogen peroxide, followed by labeling of free thiols on STING with 5-iodoacetamido-fluorescein (5-IAF) in cell lysate. We then immunoprecipitated STING protein and blotted for STING and fluorescein (FITC). Treatment with either menadione or $H_2O_2$ decreased the level of free thiols on STING, measured as decreased FITC signal relative to STING protein (*Figure 5A*). In addition, we treated mouse macrophages with diamide, a reagent that specifically oxidizes free thiols to form disulfide bonds (*Kosower and Kosower, 1995*). Although diamide treatment induced formation of the STING polymer, such polymers were likely not functional because STING activation requires ligand-induced conformational change (*Shang et al., 2019*). This was consistent with our observation that diamide treated macrophages failed to phosphorylate TBK1 and upregulate *Ifnb* upon DMXAA stimulation (*Figure 5B and C*). Thus, ROS oxidized free cysteine(s), thereby blocking the activation of both overexpressed human STING and endogenous mouse STING.

To identify the precise site of modification on STING, we labeled both reduced and oxidized cysteines with iodoacetamide (IAM) and N-ethylmaleimide (NEM) on STING. First, we treated mouse macrophages with vehicle control or menadione. This was followed by alkylation of free thiols by IAM, labeling reduced cysteines ($Cys_{red}$). We then immunoprecipitated STING, reduced oxidized thiols, and alkylated the DTT-reduced thiol groups with NEM, labeling oxidized cysteines ($Cys_{ox}$). We lastly quantified the ratio of IAM (identified as Carbamidomethylation, CAM)/NEM modification on specific cysteines using liquid chromatography with tandem mass spectrometry (LC-MS/MS) (*Figure 5D*; *Wu et al., 2020*). By this effort, we determined that C147 of endogenous murine STING was modified by both CAM and NEM (*Figure 5—figure supplement 1*). While very low levels of NEM modification ($Cys_{ox}$) were detected on C147 of endogenous murine STING in vehicle-treated sample, approximately one quarter of C147 was modified by NEM with menadione treatment, indicating increased oxidation of C147 (*Figure 5E and F*). This cysteine residue is highly conserved across all mammalian species (*Figure 5—figure supplement 2*) and is critical for STING function. C147 in mouse is equivalent to C148 in human STING. A C148A mutant of hSTING is unable to form an intermolecular disulfide bond and induce interferon response upon stimulation (*Ergun et al., 2019*). To confirm the critical role C148 in STING polymer formation, we treated 293 T cells overexpressing a C148A mutant of hSTING with diamide. While diamide induced formation of polymer on WT STING, we observed significantly less polymer formation with diamide on the C148A mutant (*Figure 5G*). Altogether, we propose that C147 on murine STING and C148 on human STING are oxidized by ROS, thereby blocking STING activation.

## ROS regulate MHV68 replication in vivo

We have shown in vitro that ROS regulate interferon induction; however, whether this regulation is physiologically important remains to be investigated. To address this question, we either induced ROS with menadione or inhibited endogenous ROS with NAC in mice. We then infected mice with a luciferase tagged-MHV68 reporter virus and monitored acute replication of the virus by luciferase signal (*Figure 6A*; *Hwang et al., 2008*; *Reese et al., 2014*). Treatment of mice with 10 mg/kg

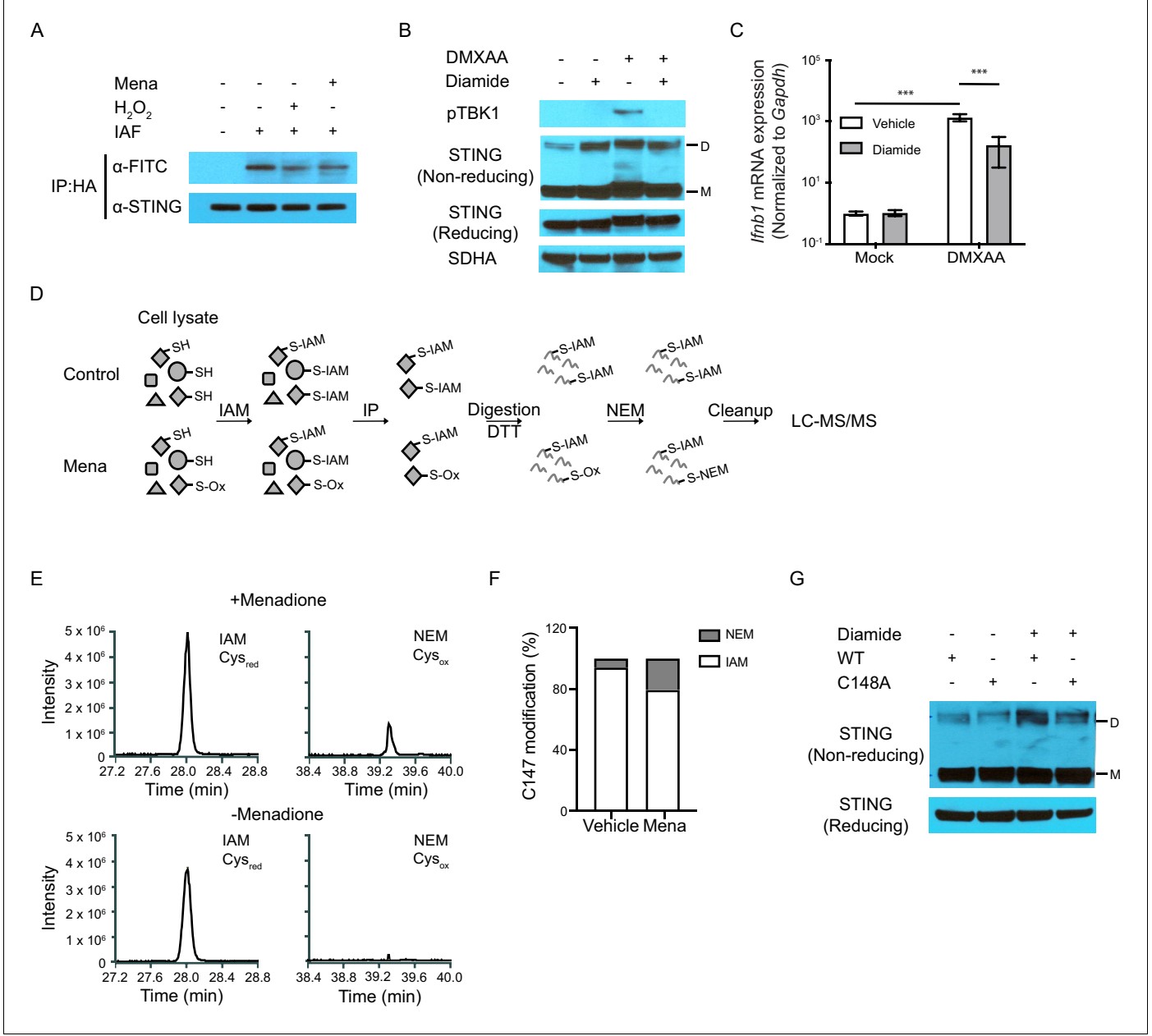

**Figure 5.** ROS oxidize C148 on STING. (**A**) *Sting-/-* fibroblasts stably expressing HA-tagged human STING were treated with vehicle or 200 μM $H_2O_2$ in serum free medium for 10 min. Cell lysates were incubated with 5 μM 5-IAF for 1 hr at room temperature to label free thiols. Protein levels of STING and FITC were detected after immunoprecipitation for HA-tagged protein. Data shown are representative results of two independent experiments. (**B, C**) BMDMs were treated with 200 μM diamide for 30 mins. STING polymers (B, n = 2) and *Ifnb* transcripts (C, n = 4) were determined at 1 hr after 1 μg/ml DMXAA stimulation. M: STING monomer; D: STING dimer. Bars represent the mean ± SE, *p* value was calculated using two-way ANOVA followed by Tukey's multiple comparison test. Only the *p* values for the most relevant comparison are shown for clarity purpose. ***, p<0.001. (**D**) Schematic of differential alkylation (IAM labeling followed by DTT reducing and NEM labeling) of cysteines for mass spectrometry analysis. (**E**) Mass spectra of IAM- and NEM-modified STING in vehicle and menadione treated samples. n = 1 (**F**) Quantification of $Cys_{red}$ and $Cys_{ox}$ from mass spectrometry analysis. (**G**) Vectors with WT STING or C148A mutated STING were transfected into HEK293T cells. Twenty-four hours after transfection, cells were treated with vehicle or 200 μM diamide for 30 min. Polymer of STING was determined with non-reducing SDS-PAGE. M: STING monomer; D: STING dimer. n = 1. The online version of this article includes the following figure supplement(s) for figure 5:

**Figure supplement 1.** Representative MS/MS spectra of SAVC$_{147}$EEK peptide in STING protein modified by both IAM (CAM) and NEM.
**Figure supplement 2.** Sequence alignment of STING from multiple species suggested C148 of STING is highly conserved across mammalian species.

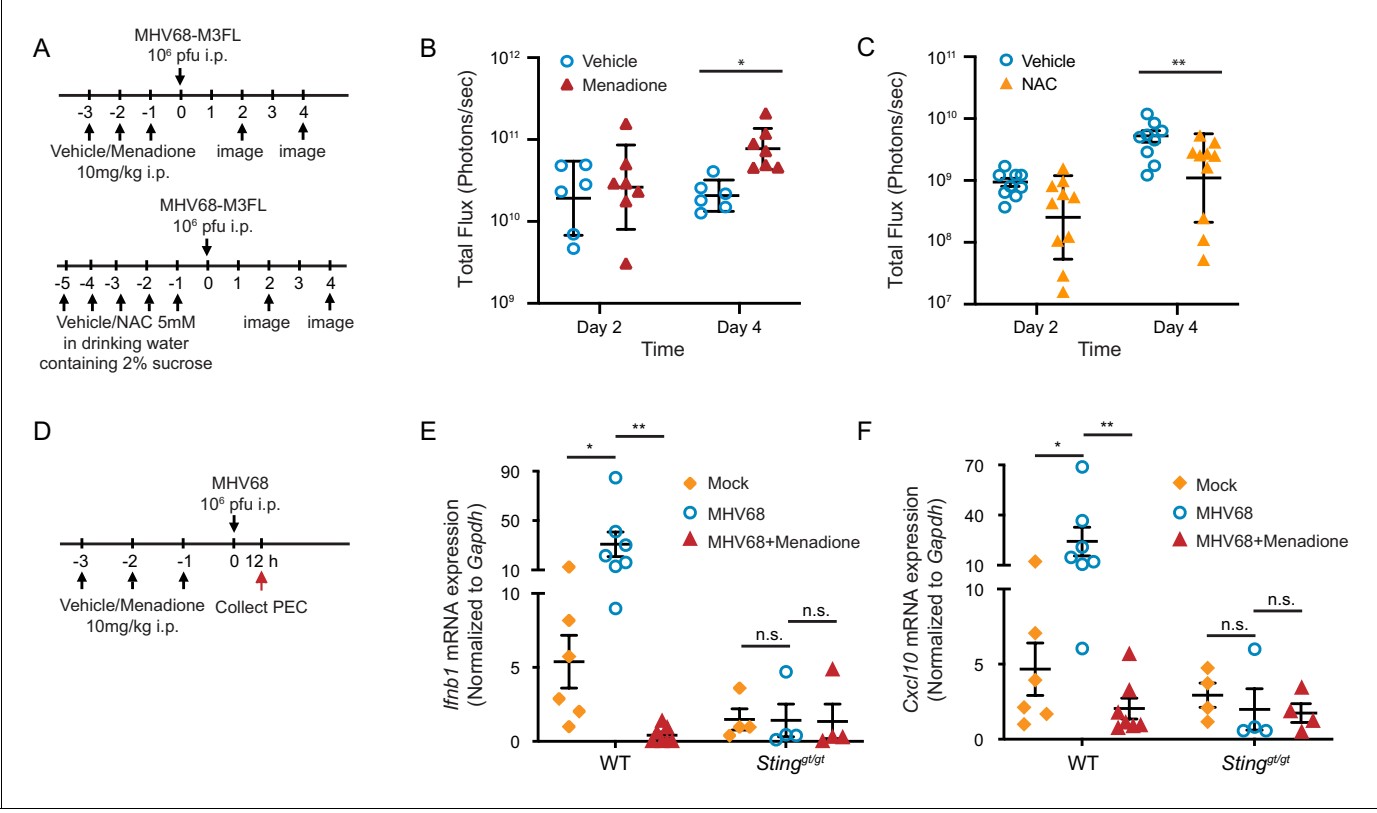

**Figure 6.** ROS regulate MHV68 replication in vivo. (A) Schematic of MHV68 replication in mice with menadione treatment or NAC treatment. (B, C) 8–12 weeks old mice were sex-matched and randomly assigned to groups prior to experiment. Mice were treated and infected as shown in (A). Total flux (photons/second) was measured using IVIS bioluminescence imager at day 2 and day 4 after infection to quantitate lytic replication of MHV68. Data shown were the results obtained from a pool of two independent experiments. Bars represent geometric mean ±geometric SD. Each dot represents an individual mouse. (D) Schematic for the quantification of interferon and ISG transcripts in mice with menadione treatment during MHV68 infection. 7–12 weeks old mice were sex-matched and randomly assigned to groups prior to experiment. Mice were treated with either vehicle control (5% DMSO in corn oil) or menadione (10 mg/kg in corn oil) for 3 days, then infected with $10^6$ PFU MHV68 peritoneally. Twelve hours post infection, transcripts of *Ifnb* and *Cxcl10* in peritoneal exudate cells (PECs) were determined from individual mice. (E, F) Transcripts of *Ifnb* and *Cxcl10* in PECs from mock (vehicle treatment, no infection), MHV68 (vehicle treatment, MHV68 infected) and MHV68+Menadione (Menadione treatment, MHV68 infected) mice. Bars represent mean ± SE, each dot represent an individual mouse. Data are collected from one controlled experiment. Statistical analysis was conducted using two-way ANOVA (repeated measures). *p<0.05.

menadione increased virus replication (*Figure 6B*). In contrast, treatment of mice with NAC inhibited virus replication (*Figure 6C*). Additionally, we measured the interferon response in both WT and *Sting*[gt/gt] mice during MHV68 infection (*Figure 6D*). While expression of *Ifnb1* and *Cxcl10* was increased in peritoneal exudate cells 12 hr after virus infection in WT mice, such response was diminished in *Sting*[gt/gt] mice. Consistent with our in vitro data, menadione treatment dramatically inhibited interferon and ISG production induced during MHV68 infection (*Figure 6E and F*). These data suggest that ROS are important for controlling virus replication in vivo.

## Discussion

Recent work indicates that ROS regulate cellular defense pathways, including Toll-like receptor signaling and inflammasome activation (*Bulua et al., 2011*; *Schieber and Chandel, 2014*; *Soucy-Faulkner et al., 2010*; *Tschopp and Schroder, 2010*; *West et al., 2011*). However, the function of ROS in DNA sensing pathways has not been investigated. Here, we found that ROS regulate cytoplasmic DNA sensing by inhibiting STING activity during herpesvirus infection. ROS suppressed the activation of STING, as well as the activation of TBK1 and IRF3, thus leading to reduced IFNβ

transcription. Importantly, we also determined that ROS oxidized a particular cysteine residue, C147, of murine STING. When we mutated the equivalent human C148 residue, diamide was no longer able to oxidize STING and form functional polymers, suggesting that redox regulation of STING required this cysteine residue. We also explored the functional consequence of ROS regulation of STING activity in vivo and demonstrated that treatment of mice during herpesvirus infection with either an inducer of ROS or a neutralizer of ROS, increased or decreased MHV68 replication in mice, respectively. Consistent with these findings, a ROS inducer inhibited interferon production during herpesvirus infection in a STING dependent manner. Our work suggests that redox modification of STING is an important mechanism for regulating STING activity, which may explain how herpesviruses manipulate STING signaling and reduce interferon levels to support their replication. Notably, elevated ROS levels in the context of viral infection is of high clinical relevance, as ROS is increased in elderly people as well as in many diseases, including cardiovascular diseases, lung fibrosis, diabetes mellitus and cancer (*Kurundkar and Thannickal, 2016*; *Ma et al., 2013*; *Ma et al., 2013*; *Di Pietro et al., 2017*; *Saadatian-Elahi et al., 2020*). This suggests that antiviral responses in these individuals could be repressed, leading to impaired control of chronic herpesvirus infections and poor clinical outcomes (*Stowe et al., 2007*; *Ye et al., 2016*).

Structural and functional data prior to our study indicate that STING is oxidized under certain circumstances, but the precise site(s) of oxidation and the role of ROS in STING regulation has been unclear. On one hand, nuclear factor erythroid 2-related factor 2 (NRF2), which drives the expression of antioxidant genes, decreases STING-induced interferon and thus increases susceptibility to herpes simplex virus-2 (HSV-2) (*Gunderstofte et al., 2019*; *Olagnier et al., 2018*). These studies did not indicate whether STING itself is oxidized in the presence or absence of NRF2. On the other hand, induction of oxidative stress by the complex I inhibitor rotenone inhibits ectopically expressed STING activity and interferon production (*Jin et al., 2010*).

The malleable chemistry of cysteines makes them optimal targets for redox regulation (*Bindoli et al., 2008*; *Wang et al., 2012*). Cysteines are susceptible to multiple post-translational modifications, including sulfenylation, SOH; sulfinylation, $SO_2H$; sulfinylation, $SO_3H$; glutathionylation, -SSG; protein disulfide formation; nitrosylation, etc., all of which can affect protein structure and function. Moreover, many cysteine oxidative modifications are reversible, making them ideal for initiating and terminating signals. Redox modification can also be either activating or inhibiting for protein function. For example, sulfenylation of the EGFR catalytic site enhances kinase activity whereas oxidation-induced disulfide bond formation inactivates MKK6 (*Paulsen et al., 2012*; *Wani et al., 2011*).

Previous reports indicate that cysteine residues of STING may form disulfide-containing polymers (*Ergun et al., 2019*; *Jin et al., 2010*; *Jönsson et al., 2017*; *Li et al., 2015*; *Motani et al., 2015*). A recent report identified redox regulation of C206 upon cGAMP binding to STING (*Cuervo et al., 2020*). The authors also indicate the C148 may be oxidized under baseline conditions but did not characterize this oxidation. In our report, we identify C148 oxidation of STING as a posttranslational modification that negatively regulates STING activation. Our model suggests that increased ROS promote oxidation of C148. C148 is important for forming disulfide bridges between STING dimers, leading to the formation of stabilized polymers in the ER (*Ergun et al., 2019*; *Shang et al., 2019*). When C148 is oxidized, STING no longer forms stable polymers, thereby preventing recruitment and activation of TBK1 (*Figure 7*; *Zhang et al., 2019*). Notably, our data do not distinguish whether oxidized STING can still bind to cGAMP or if oxidation prevents cGAMP binding and conformational change of STING dimers.

Some pathogens may have evolved to take advantage of ROS to inhibit cellular defenses. Indeed, multiple herpesviruses induce oxidative stress in infected cells and ROS promote virus replication in vitro. Using a mouse model gammaherpesvirus, MHV68, we demonstrated that in vivo treatment of mice with menadione to induce ROS increased virus replication and treatment of mice with NAC to inhibit ROS suppressed virus replication. The effects of menadione and NAC treatments were relatively modest in these in vivo experiments. However, our results are consistent with previous reports that PRRs other than STING also participate in the control of MHV68 infection (*Bussey et al., 2019*; *Sun et al., 2015*). In addition, another group showed that implantation of KSHV-infected cells into mice and treatment with NAC inhibits lytic replication of this human gammaherpesvirus (*Ye et al., 2011*). Further work is required to determine if ROS inhibition of cytoplasmic DNA sensing promotes

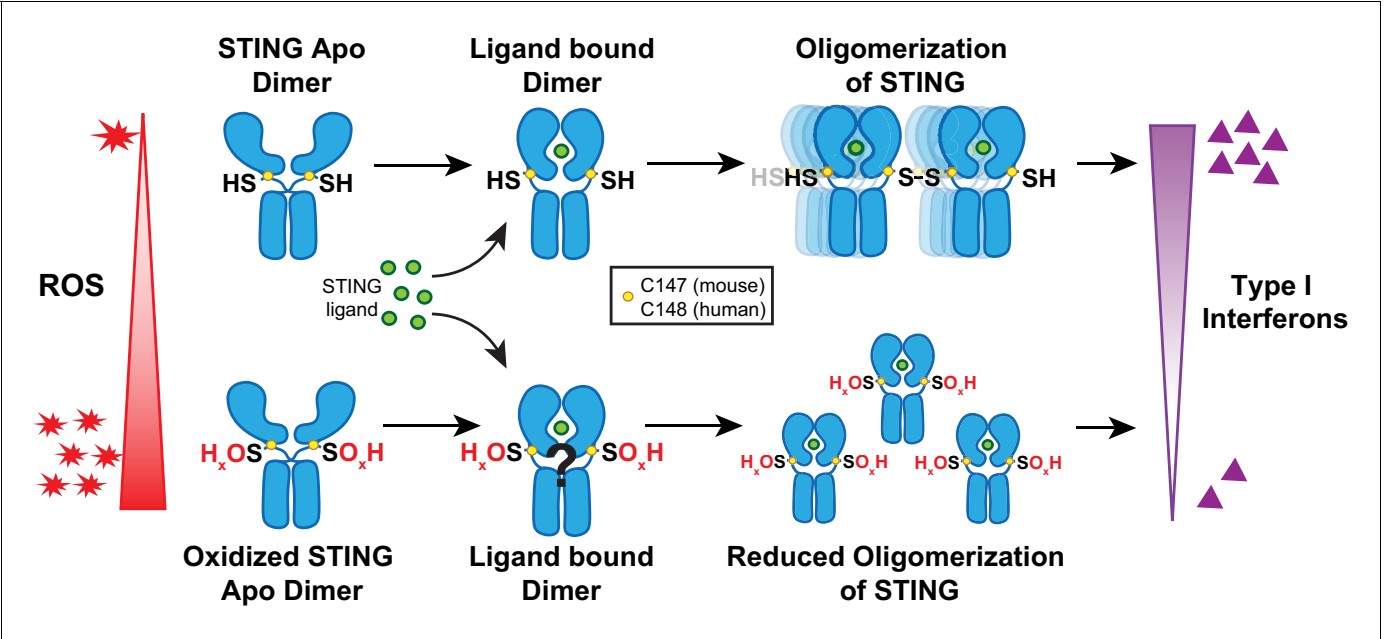

**Figure 7.** Model of ROS modification of STING. Increasing ROS leads to oxidation of free thiol on C148 of human STING and C147 of mouse STING. This prevents polymerization of STING upon binding of 2',3'-cGAMP and leads to reduced interferon production.

reactivation of herpesviruses from latency in vivo and whether particular herpesvirus-encoded genes alter the ROS levels in the cell.

Consistent with our data that STING undergoes redox modification, a recent study demonstrates that glutathione peroxidase 4 (GPX4), an enzyme that protects cells against membrane lipid peroxidation and maintains redox homeostasis, is required for the activation of the cGAS-STING pathway. This report also shows that GPX4 is required for innate immune responses against herpes simplex virus 1 (*Jia et al., 2020*). These data, along with our data, support a model whereby ROS promote DNA virus acute replication in a STING-dependent manner.

Oxidative inhibition of STING may also be important for preventing overactivation of the interferon response due to other sources of cytosolic DNA, not just viral infection. Mitochondrial damage and release of mtDNA into the cytoplasm activates the cGAS-STING pathway (*Chen et al., 2018*; *McArthur et al., 2018*; *West et al., 2015*). In the absence of cellular caspases, mtDNA induces type I interferon production (*White et al., 2014*). Notably, many of the stimuli that lead to mitochondrial fragmentation and release of mtDNA also produce mitochondrial ROS (*Araujo et al., 2018*; *Willems et al., 2015*; *Woo et al., 2012*). Redox regulation of STING may represent a negative regulatory mechanism to limit aberrant production of type I interferon.

Multiple autoimmune and autoinflammatory diseases are associated with increased production of interferon and are termed type I interferonopathies. These monogenic diseases are driven by mutations in DNases or STING (*Motwani et al., 2019*). Particularly, the rare autoinflammatory disease called STING-associated vasculopathy with onset in infancy (SAVI) is caused by gain-of-function mutations in the gene that encodes STING, *TMEM173* (*Liu et al., 2014*). Of the six patients identified with SAVI, all had point mutations in close proximity to C148, with one patient having a V147L mutation. It is therefore plausible that this region is important for negatively regulating STING activity and normalizing the 'basal' or tonic levels of interferon in cells (*Gough et al., 2012*). We found that NAC treatment of macrophages prior to viral infection increased *Ifnb* transcript levels but did not significantly alter MHV68 growth. The small change in basal ROS with NAC treatment may be insufficient to suppress virus replication. However, changes in the low level of endogenous ROS in cells could regulate basal interferon through STING activation and could contribute interferonopathies over the long term.

Altogether, our results identify a critical post-translational modification of STING and indicate that redox regulation of STING is important for innate immune responses against certain DNA viruses. The cysteine at position 147 in mice and 148 in humans is a critical cysteine for STING polymer formation and activation. Identifying the post-translational modifications of STING contributes not only to our understanding of the basic biology of the cGAS/STING pathway but is also critical for identifying novel immunotherapies to target interferon production.

# Materials and methods

**Key resources table**

| Reagent type (species) or resource | Designation | Source or reference | Identifiers | Additional information |
|---|---|---|---|---|
| Strain, strain background (*Mus musculus*) | C57BL/6J | The Jackson Laboratory | Stock No. 000664 | Bred at UTSW facility with IACUC approval |
| Strain, strain background (*Mus musculus*) | C57BL/6J-*Tmem173$^{gt}$*/J | The Jackson Laboratory | Stock No. 017537 | Bred at UTSW facility with IACUC approval |
| Strain, strain background (*Mus musculus*) | B6.129S2-*Ifnar1$^{tm1Agt}$*/J | The Jackson Laboratory | Stock No. 32045 | Bred at UTSW facility with IACUC approval |
| Strain, strain background (*Mus musculus*) | B6.129P2-*Acox1$^{tm1Jkr}$*/J | The Jackson Laboratory | Stock No. 029747 | Bred at UTSW facility with IACUC approval |
| Cell line (*Herpesviridae, Rhadinovirus*) | Murine gamma herpesvirus 68 (WUSM stain) | ATCC | VR-1465 | |
| Cell line (*Herpesviridae, Rhadinovirus*) | Murine gamma herpesvirus 68-M3FL | Home made *Hwang et al., 2008* | | |
| Cell line (*Mus musculus*) | 3T12 | ATCC | Cat# ATCC CCL-164; RRID:CVCL_0637 | |
| Cell line (*Homo sapiens*) | 293T | ATCC | Cat# ATCC CRL-3216; RRID:CVCL_0063 | |
| Transfected construct (*Homo sapiens*) | STING | GenBank | AVQ94753.1 | Express STING into 293T cells |
| Antibody | Anti-STING (Rabbit polyclonal) | Proteintech | Cat# 19851-1-AP; RRID:AB_10665370 | WB (1:1000); IP (1 ug/ml) |
| Antibody | Anti-STING (Rabbit monoclonal) | Cell Signaling | Cat# 50494S; RRID:AB_2799375 | WB (1:1000) |
| Antibody | Anti-TBK1/NAK (Rabbit monoclonal) | Cell Signaling | Cat# 3504S; RRID:AB_2255663 | WB (1:1000) |
| Antibody | Anti-IRF-3 (Rabbit monoclonal) | Cell Signaling | Cat# 4302S; RRID:AB_1904036 | WB (1:1000) |
| Antibody | Anti-Phospho-TBK1 (Ser172) (Rabbit monoclonal) | Cell Signaling | Cat# 5483S; RRID:AB_10693472 | WB (1:1000) |
| Antibody | Anti-Phospho-IRF-3 (Ser396) (Rabbit monoclonal) | Cell Signaling | Cat# 4947S; RRID:AB_823547 | WB (1:1000) |
| Antibody | Mouse monoclonal anti-SDHA antibody | Abcam | Cat# ab14715; RRID:AB_301433 | WB (1:5000) |
| Antibody | Anti-MHV68 (Rabbit polyclonal) | Home made *Weck et al., 1997* | | FACs (1:1000) |

*Continued on next page*

*Continued*

| Reagent type (species) or resource | Designation | Source or reference | Identifiers | Additional information |
|---|---|---|---|---|
| Antibody | Anti-Rabbit IgG(H+L) secondary antibody, Alexa Fluor 647 (Goat polyclonal) | Thermo Fisher Scientific | Cat# A-21245; RRID:AB_2535813 | FACs (1:4000) |
| Antibody | Anti-Rabbit IgG, Peroxidase (Donkey polyclonal) | Jackson Immuno Research Laboratory | Cat# 711-035-152; RRID:AB_10015282 | WB (1:5000) |
| Antibody | Anti-Mouse IgG, Peroxidase (Goat polyclonal) | Jackson Immuno Research Laboratory | Cat# 115-035-174; RRID:AB_2338512 | WB (1:5000) |
| Antibody | Anti-FITC (Rabbit polyclonal) | Thermo Fisher Scientific | Cat# 71-1900; RRID:AB_2533978 | WB (1:2000) |
| Recombinant DNA reagent | pcDNA 3.1(+) Mammalian Expression Vector (plasmid) | Thermo Fisher Scientific | Cat# V79020 | Vector for the expression of human STING |
| Sequence-based reagent | *Ifnb* forward | This paper | qPCR primers | CAGCTCCAAGAAAGGACGAAC |
| Sequence-based reagent | *Ifnb* reverse | This paper | qPCR primers | GGCAGTGTAACTCTTCTGCAT |
| Sequence-based reagent | *Cxcl10* forward | This paper | qPCR primers | TTAACGTCAGGCCAACAGAG |
| Sequence-based reagent | *Cxcl10* reverse | This paper | qPCR primers | GAGGGAAACCAGGAAAGATAGG |
| Sequence-based reagent | *Isg15* forward | This paper | qPCR primers | CAGGACGGTCTTACCCTTTCC |
| Sequence-based reagent | *Isg15* reverse | This paper | qPCR primers | AGGCTCGCTGCAGTTCTGTAC |
| Sequence-based reagent | *Isg20* forward | This paper | qPCR primers | CCATGGACTGTGAGATGGTG |
| Sequence-based reagent | *Isg20* reverse | This paper | qPCR primers | CTCGGGTCGGATGTACTTGT |
| Sequence-based reagent | *Gapdh* forward | This paper | qPCR primers | GGGTGTGAACCACGAGAAATA |
| Sequence-based reagent | *Gapdh* reverse | This paper | qPCR primers | GTCATGAGCCCTTCCACAAT |
| Sequence-based reagent | *Gsr* forward | This paper | qPCR primers | CACCGAGGAACTGGAGAATG |
| Sequence-based reagent | *Gsr* reverse | This paper | qPCR primers | ATCTGGAATCATGGTCGTGG |
| Sequence-based reagent | *Gclm* forward | This paper | qPCR primers | AATCAGCCCCGATTTAGTCAG |
| Sequence-based reagent | *Gclm* reverse | This paper | qPCR primers | CGATCCTACAATGAACAGTTTTGC |
| Sequence-based reagent | C148 forward | This paper | Site direct mutagenesis PCR primers | CTCTGCAGTGCTGAAAA AGGGAATTTCAACGTGGC |
| Sequence-based reagent | C148A reverse | This paper | Site direct mutagenesis PCR primers | ATCTCAGCTGGGGCCAGG |
| Commercial assay or kit | Lipofectamine 3000 Reagent | Thermo Fisher Scientific | Cat# L3000008 | |
| Commercial assay or kit | Qiagen RNeasy Mini Kit | Qiagen | Cat# 74104 | |

*Continued on next page*

*Continued*

| Reagent type (species) or resource | Designation | Source or reference | Identifiers | Additional information |
|---|---|---|---|---|
| Commercial assay or kit | SuperScript VILO cDNA Synthesis Kit | Thermo Fisher Scientific | Cat# 11754050 | |
| Commercial assay or kit | PowerUp SYBR Green Master Mix | Thermo Fisher Scientific | Cat# A25776 | |
| Commercial assay or kit | LIVE/DEAD Fixable Dead Cell Stain Kits | Invitrogen | Cat# L34975 | |
| Commercial assay or kit | Q5 Site-Directed Mutagenesis Kit | New England BioLabs | Cat# E0554S | |
| Chemical compound, drug | DMXAA | InvivoGen | Cat# tlrl-dmx | (1 ug/ml) for macrophages, (2 ug/ml) for fibroblasts |
| Chemical compound, drug | 2'3'-cGAMP | InvivoGen | Cat# tlrl-nacga23 | (10 ug/ml) |
| Chemical compound, drug | ISD | InvivoGen | Cat# tlrl-isdn | (10 ug/ml) |
| Chemical compound, drug | poly(dA:dT) | InvivoGen | Cat# tlrl-patn-1 | (1 ug/ml) |
| Chemical compound, drug | poly(I:C) | Invivogen | Cat# tlrl-picw | (1 ug/ml) |
| Chemical compound, drug | D-Luciferin, Potassium Salt | GOLDBIO | Cat# LUCK | (150 mg/kg) |
| Chemical compound, drug | Menadione | Sigma | Cat# M9429 | (10 mg/kg) for mice, concentration for cells were indicated in each experiment |
| Chemical compound, drug | Hydrogen peroxide solution | Sigma | Cat# 216763 | Concentration for cells were indicated in each experiment |
| Chemical compound, drug | N-Acetyl-L-cysteine | Sigma | Cat# A7250 | 2 mM for macrophages, 5 mM in drinking water with 2% sucrose |
| Chemical compound, drug | Iodoacetamide | Sigma | Cat# I1149 | 100 mM |
| Chemical compound, drug | N-ethylmaleimide | Sigma | Cat# E3876 | 50 mM |
| Chemical compound, drug | Elastase | Worthington | Cat# LS006363 | 1 mg/ml |
| Chemical compound, drug | 5-(Iodoacetamido) fluorescein | Sigma | Cat# I9271 | 5 ug/ml |
| Chemical compound, drug | Diamide | Sigma | Cat# D3648 | 200 uM |
| Software, algorithm | GraphPad Prism 8 | GraphPad | www.graphpad.com; RRID:SCR_002798 | |
| Software, algorithm | FlowJo software | FlowJo | www.flowjo.com; RRID:SCR_008520 | |
| Software, algorithm | Live Image Software | Perkin Elmer | www.perkinelmer.com; RRID:SCR_014247 | |

## Animals

C57BL/6J, C57BL/6J-*Tmem173*[gt]/J (**Sauer et al., 2011**), B6.129S2-*Ifnar1*[tm1Agt]/J (**Muller et al., 1994**), B6.129P2-*Acox1*[tm1Jkr]/J (**Fan et al., 1996**). were purchased from The Jackson Laboratory. All mice were housed in a specific pathogen-free, double-barrier facility at the University of Texas

Southwestern Medical Center. Mice were maintained and used under a protocol approved by UT Southwestern Medical Center Institutional Animal Care and Use Committee (IACUC).

## Chemicals

Menadione, hydrogen peroxide solution, diamide and N-Acetyl-L-Cysteine were purchased from Sigma-Aldrich. Iodoacetamide (IAM), N-Ethylmaleimide (NEM), 5-iodoacetamido-fluorescein (5-IAF) were purchased from Sigma-Aldrich to label free thiols on protein. cGAMP, ISD, DMXAA, poly I:C, poly dA:dT were purchased from Invivogen to induce interferon signaling.

## Chemical treatment

A stock solution of menadione was prepared in DMSO at 50 mg/ml, then diluted to different concentrations with culture medium. Cells were treated with menadione by replacing culture medium with fresh medium containing menadione and incubated at 37°C for a duration indicated in each experiment. A stock solution of hydrogen peroxide was prepared in PBS at the concentration of 1 M, then diluted to different concentration with serum-free medium to avoid the decomposition of hydrogen peroxide by residual catalases in FBS. However, the treatment of hydrogen peroxide was conducted in culture medium containing FBS for the flow cytometric quantification of viral growth in BMDMs, in which case, serum is essential to support virus growth.

## Cell culture

Bone marrow derived macrophages were differentiated in DMEM (Corning) with 10% FBS (Biowest) supplemented with 1% glutamine (Corning), 1% HEPES (Corning) and 10% CMG14 (*Takeshita et al., 2000*) supernatant for 7 days. 3T12 cells (ATCC, CCL-164, mycoplasma tested) were maintained in DMEM with 5% FBS supplemented with 1% glutamine and 1% HEPES. 293 T cells (ATCC, CRL-3216, mycoplasma tested), fibroblasts overexpressing human STING were maintained in DMEM with 10% FBS. Primary fibroblasts (MEFs) were isolated from embryonic tissue in DMEM with 10% FBS supplemented with 1% glutamine, 1% HEPES, then passed and maintained in the same culture medium for further propagation.

## Generation of virus stocks

Murine γ-herpesvirus 68 (WUSM stain) was purchased from ATCC. Murine γ-herpesvirus 68-M3FL was generated as previously reported (*Hwang et al., 2008*). Virus stock was generated in 3T12 cells and aliquots of virus were stored at −80°C.

## Virus infection

Fully differentiated BMDMs were seeded on 24 well plates ($1.5 \times 10^5$ cells per well) or six well plates ($10^6$ cells per well). Cells were pretreated with mock control or menadione at 8 μM, 4 μM, or 2 μM for 16 hr. The next day, macrophages were infected with MHV68 at MOI = 5. After an hour, cells were washed with PBS twice to remove unabsorbed virus and resuspended in medium with or without treatments. For the viral growth curve, samples were collected at 0 hr, 24 hr, 48 hr, 72 hr and 96 hr after infection and were frozen at −80°C. The titer of virus was determined by plaque assay in 3T12 cells. For qRT-PCR, cells were washed with ice-cold PBS twice at 6 hr after infection and were frozen at −80°C.

## Plaque assay

The concentration of virus was quantitated by plaque assay in 3T12 cells. The frozen samples containing virus were thawed in an incubator at 37°C. The samples were serial diluted, then added to a monolayer of 3T12 cells. After an hour of absorption, the cells were then covered with 1% methylcellulose. Plates were incubated at 37°C for 7 days, and the monolayers were stained with 0.1% crystal violet.

## Flow cytometry for MHV68 lytic protein positive cells

To determine the percentage of cells that express lytic proteins of MHV68 infection, cells were harvested 24 hr after infection and fixed with 2% formaldehyde. The cells were blocked with 10% mouse serum and 1% Fc block (anti-CD16/32, clone 2.4G2, Tonbo), and then stained with polyclonal

rabbit antibody to MHV68 (1:1000) (*Reese et al., 2014*; *Weck et al., 1997*), followed by secondary goat anti-rabbit Alexa Fluor-647 (Thermo Fisher, Invitrogen, A-21244).

## Cell viability assay

Fully differentiated BMDMs were treated with different concentrations of menadione as indicated. Cells were then scraped and collected at different time points and stained with LIVE/DEAD Fixable Near-IR Dead Cell Stain Kit (ThermoFisher Scientific) for 30 min at room temperature in the dark. After washing with PBS, dead cells were identified using flow cytometry.

## Transfection

293 T cells were seeded on six well plates. The next day, cells were transfected with WT STING vector or C148A STING vector using Lipofectamine 3000 (Thermo Fisher Scientific) according to the manufacturer's protocol.

## Western blot

Cells were lysed with RIPA buffer (150 mM NaCl, 1% NP-40, 0.5% sodium deoxycholate, 0.1% SDS, 25 mM Tris with protease inhibitor cocktail (Roche)). Protein concentrations were determined using the Bradford assay (Bio-Rad). Equal amounts of protein were mixed with 5x loading sample buffer containing 2-Mercaptoethanol and heated at 97°C for seven mins. The samples were resolved by 4–12% Bis-Tris plus gels (Thermo Fisher Scientific) and transferred to a nitrocellulose membrane. Proteins were labeled with primary antibodies against STING (1:1000, Catalogue no.13647S, Cell Signaling; 1:1000, Catalogue no. 1985–1-AP, Proteintech), TBK1 (1:1000, Catalogue no. 3504S, Cell Signaling), IRF3 (1:1000, Catalogue no. 4302S, Cell Signaling), pTBK1 (1:1000, Catalogue no. 5483S, Cell Signaling), pIRF3 (1:1000, Catalogue no. 4947S, Cell Signaling), SDHA (1:5000, Catalogue no. ab14715, Abcam), β-actin (1:5000, Catalogue no. A2228, Sigma). Secondary antibodies used were donkey-anti-rabbit (1:5000, Catalogue no.711-035-152, Jackson ImmunoResearch Laboratory) and goat-anti-mouse peroxidase (1:5000, Catalogue no.115-035-174 Jackson ImmunoResearch Laboratory). Membranes were developed using Luminata Forte Western HRP substrate (Millipore). For non-reducing SDS-PAGE, protein was mixed with 5x loading sample buffer and incubated at RT for 1 hr without boiling. Samples were then resolved by 4–12% Bis-Tris plus gels.

## RT-qPCR

Cells were plated in six well plates, either infected with MHV68 or induced with STING ligand as indicated. RNA was extracted using RNeasy Mini Kit (Qiagen) and reverse transcribed into cDNA using SuperScript VILO cDNA Synthesis Kit (Thermo Fisher Scientific). Relative quantification of target genes was determined using PowerUp SYBR Green Master Mix (Thermo Fisher Scientific) in a QuantStudio 7 Flex real time PCR system. Sequences of primers are as follow:

| Primers | Sequence |
| --- | --- |
| *Ifnb* forward | CAGCTCCAAGAAAGGACGAAC |
| *Ifnb* reverse | GGCAGTGTAACTCTTCTGCAT |
| *Cxcl10* forward | TTAACGTCAGGCCAACAGAG |
| *Cxcl10* reverse | GAGGGAAACCAGGAAAGATAGG |
| *Isg15* forward | CAGGACGGTCTTACCCTTTCC |
| *Isg15* reverse | AGGCTCGCTGCAGTTCTGTAC |
| *Isg20* forward | CCATGGACTGTGAGATGGTG |
| *Isg20* reverse | CTCGGGTCGGATGTACTTGT |
| *Gapdh* forward | GGGTGTGAACCACGAGAAATA |
| *Gapdh* reverse | GTCATGAGCCCTTCCACAAT |
| *Gsr* forward | CACCGAGGAACTGGAGAATG |
| *Gsr* reverse | ATCTGGAATCATGGTCGTGG |
| *Gclm* forward | AATCAGCCCCGATTTAGTCAG |
| *Gclm* reverse | CGATCCTACAATGAACAGTTTTGC |

## Immunoprecipitation

Cells were lysed with gentle lysis buffer (Cell Signaling) for 15 mins, and then the cell lysate was spun down at 16,000 g for 10 mins. STING antibody (Proteintech) was added into the cell lysate at 1 µg/mL and incubated overnight at 4°C with rotation. The next day, 20 µL of Pierce protein A/G magnetic beads (Thermo Fisher Scientific) were added into each sample and incubated at 4°C for 2 hr to capture STING protein. Samples were then washed with lysis buffer five times, and protein was dissociated from beads by heating at 97°C for seven mins.

## STING cloning

FLAG-tagged STING was cloned into pcDNA 3.1 (+) mammalian vector. C148A mutant was generated by site-directed mutagenesis (New England Biolabs). Vectors with wildtype STING and C148A mutated STING were transfected into HEK293T cells for transient STING overexpression. Mutagenesis primers: forward-CTCTGCAGTGCTGAAAAAGGGAATTTCAACGTGGC; reverse-ATCTCAGC TGGGGCCAGG.

## Detection of oxidative modification on STING

*Sting-/-* fibroblasts stably expressing HA-tagged human STING were plated on 10 cm dishes at a density of $10^7$ cells per dish. Twenty-four hours later, growth medium with 10% FBS was replaced with DMEM supplemented with 2% FBS. The next day, cells were treated with mock control, 25 µM menadione for 30 mins or 200 µM $H_2O_2$ for 10 mins in serum-free DMEM. Cells were then washed with ice cold PBS twice and lysed with gentle cell lysis buffer (Cell signaling) containing 5 µM 5-IAF which labels free thiols with a fluorescein (FITC) tag (*Ostman et al., 2011*). Cell lysate was spun down at 16,000 g for 10 mins, and the supernatant was then incubated at RT for 1 hr in the dark. Lysates were subjected to immunoprecipitation for HA-tagged STING and probed for both STING and FITC using western blot.

## Mass spectrometry analysis of STING cysteine oxidation

Quantification of cysteine oxidation on STING with mass spectrometry was completed as described previously (*Wu et al., 2020*). Differentiated BMDMs were treated with serum free medium or 25 µM menadione for 30 mins, and proteins were extracted using gentle cell lysis buffer (Cell signaling) with 100 mM iodoactamide to label free thiols ($Cys_{red}$). After immunoprecipitation, the STING-Trap beads were incubated with alkylation buffer (100 mM iodoacetamide, 2% SDS and 150 mM Tris, PH 8.0) at room temperature for 1 hr to sufficiently label free thiols on STING. Proteins were separated by SDS-PAGE, and the bands corresponding to STING were excised. The protein gel band was digested overnight with elastase (Worthington) following reduction with DTT and a second alkylation step with N-ethylmaleimide to label oxidized thiols ($Cys_{ox}$). The samples then underwent solid-phase extraction cleanup with an Oasis HLB µElution plate (Waters) and the resulting samples were analyzed by LC-MS/MS, using an Orbitrap Fusion Lumos mass spectrometer (Thermo Electron) coupled to an Ultimate 3000 RSLC-Nano liquid chromatography system (Dionex). Raw MS data files were converted to a peak list format and analyzed using the central proteomics facilities pipeline (CPFP), version 2.0.3 (*Trudgian et al., 2010*; *Trudgian and Mirzaei, 2012*). Peptide identification was performed with a non-specific enzyme search using the Open MS Search Algorithm (OMSSA) (*Geer et al., 2004*) search engine against the mouse protein database from UniProt, with common contaminants and reversed decoy sequences appended (*Elias and Gygi, 2007*). Fragment and precursor tolerances of 10 ppm and 0.5 Da were specified, and three missed cleavages were allowed. Oxidation of Met and carbamidomethylation (iodoacetamide modification) and N-ethylmaleimide modification of Cys were set as variable modifications. Mass spectrometry data were deposited online, with the link of: http://massive.ucsd.edu/ProteoSAFe/status.jsp?task= 03fb020e0fe9474ea5fa9326219f7cee.

## MHV68 acute replication in mouse

Experiments were carried out using 8–12 weeks old mice under the protocol approved by IACUC. Mice were sex-matched and randomly allocated into groups prior to experiments. For menadione

treatment, mice were injected intraperitoneally with either vehicle control (5% DMSO in corn oil) or menadione (10 mg/kg) for 1 week, starting 3 days before virus infection. For NAC treatment, 5 mM NAC was provided in the water with presence of 2% sucrose that mice were allowed to drink ad libitum throughout the experimental period starting 5 days before infection. Mice were then infected with MHV68-M3FL with a dose of $10^6$ PFU by intraperitoneal injection. To quantify virus-encoded luciferase expression (*Hwang et al., 2008*). Mice were weighed and injected with 150 mg/kg of D-Luciferin (GOLDBIO) prior to imaging using an IVIS Lumina III In Vivo Imaging System (PerkinElmer). Total flux (Photons/second) of the abdominal region was determined using Living Image software (PerkinElmer) by designating a circular region of interest (ROI) for each mouse.

## Measure transcripts of *Ifnb1* and *Cxcl10* in peritoneal exudate cells (PECs)

Mice aged between 8–12 weeks were sex-matched and randomly allocated into groups prior to experiments. Mice were injected intraperitoneally with either vehicle control (5% DMSO in corn oil) or menadione (10 mg/kg in corn oil) starting 3 days before infection. Mice were then infected with MHV68 at $10^6$ PFU by intraperitoneal injection. Twelve hours after infection, PECs from each mouse were collected. RNA was extracted and reverse transcribed. Transcripts of *Ifnb1* and *Cxcl10* were the quantified by qRT-PCR.

## Quantification and statistical analysis

Bars are mean ± SE unless otherwise stated in figure legend. Statistical comparisons were performed using GraphPad Prism 7.0 software. *p* value was computed using unpaired one-way or two-way ANOVA. Statistical significance was set at $p < 0.05$. The numbers of independent replicates (n) are reported in the figure legends.

## Acknowledgements

We thank members of the Reese and Yan labs for helpful discussion and technical assistance. We also thank the UTSW Flow Cytometry core and the UTSW Proteomics core for technical assistance. The Reese lab is supported by the American Heart Association (17SDG33670071), NIH (1R01AI130020-01A1), CPRIT (RP200118) and the Pew Scholars Program.

## Additional information

### Funding

| Funder | Grant reference number | Author |
| --- | --- | --- |
| National Institute of Allergy and Infectious Diseases | 1R01AI130020-01A1 | Lili Tao<br>Guoxun Wang<br>Christina Zarek<br>Alexandria Lowe<br>Tiffany A Reese |
| American Heart Association | 17SDG33670071 | Lili Tao<br>Guoxun Wang<br>Christina Zarek<br>Alexandria Lowe<br>Tiffany A Reese |
| Pew Charitable Trusts | | Tiffany A Reese |
| Cancer Prevention and Research Institute of Texas | RP200118 | Lili Tao<br>Guoxun Wang<br>Christina Zarek<br>Alexandria Lowe<br>Tiffany A Reese |

The funders had no role in study design, data collection and interpretation, or the decision to submit the work for publication.

## Author contributions
Lili Tao, Conceptualization, Data curation, Formal analysis, Validation, Investigation, Visualization, Methodology, Writing - original draft, Writing - review and editing; Andrew Lemoff, Formal analysis, Investigation, Methodology; Guoxun Wang, Investigation, Visualization, Writing - original draft, Writing - review and editing; Christina Zarek, Visualization, Writing - original draft, Writing - review and editing; Alexandria Lowe, Project administration, Writing - review and editing; Nan Yan, Conceptualization, Writing - review and editing; Tiffany A Reese, Conceptualization, Data curation, Supervision, Funding acquisition, Writing - original draft, Project administration, Writing - review and editing

## Author ORCIDs
Lili Tao https://orcid.org/0000-0002-9682-3911
Andrew Lemoff https://orcid.org/0000-0002-4943-0170
Guoxun Wang https://orcid.org/0000-0001-9825-3150
Nan Yan http://orcid.org/0000-0002-0637-3989
Tiffany A Reese https://orcid.org/0000-0003-2325-6546

## Ethics
Animal experimentation: This study was performed in strict accordance with the recommendations in the Guide for the Care and use of Laboratory Animals of the National Institute of Health. Mice were maintained and handled under a protocol approved by UT Southwestern Medical Center Institutional Animal Care and Use Committee (IACUC) protocol number 2015-100990.

## Decision letter and Author response
Decision letter https://doi.org/10.7554/eLife.57837.sa1
Author response https://doi.org/10.7554/eLife.57837.sa2

# Additional files

## Supplementary files
- Source data 1. All source data.
- Transparent reporting form

## Data availability
Mass spectrometry data is available at https://doi.org/10.25345/C5MT4W. Source data is available in Prism format.

The following dataset was generated:

| Author(s) | Year | Dataset title | Dataset URL | Database and Identifier |
|---|---|---|---|---|
| Reese T | 2020 | Reactive Oxygen Species Oxidize STING and Suppress Interferon Production | https://doi.org/10.25345/C5MT4W | MassIVE UCSD, 10.25345/C5MT4W |

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
