## [Decision Letter]

**Acceptance summary:**

Tao et al. followed up on an earlier identified phenomenon that reactive oxygen species (ROS) do not only inhibit pathogen growth, but can also promote replication of certain DNA viruses. The study describes the molecular mechanisms that drive ROS-dependent virus replication. Using MHV68 as model system the authors show that ROS impairs interferon response during infection and that the inhibition occurs downstream of cytoplasmic DNA sensing. Moreover, they identify that ROS suppresses the type I interferon response by oxidation of Cysteine-147 of mouse STING/Cysteine-148 of human STING. Collectively, the results of this study demonstrate that redox modification of STING is an important regulatory mechanism for STING activity during viral infection.

**Decision letter after peer review:**

Thank you for submitting your manuscript "Reactive Oxygen Species Oxidize STING and Suppress Interferon Production" to *eLife*. Your manuscript was evaluated by three reviewers, and the evaluation has been overseen by a Reviewing Editor and a Senior Editor. The Reviewing Editor has drafted this decision to help you prepare a revised submission.

All three reviews are generally supportive of your results, but they raised several questions that need to be addressed in revised manuscript. Although reviewers did not explicitly ask for additional experiments, you are welcome to include new data if you feel they would be helpful in addressing reviewers' questions.

Below please find the summary of concerns raised by reviewers.

The presented findings are novel and relevant for the readers of *eLife*. The experiments are overall well performed and the data seem scientifically sound. The manuscript is well written and the conclusions are supported by the data, but some conclusions need clarification.

Essential revisions:

1) As mentioned by the authors, herpesviruses have been described to induce ROS in phagocytic cells. In multiple figures the authors show that Menadione increases virus replication in multi-step growth curves in BMDM, indicating that virus-induced ROS does not reach maximum levels. Unclear is, whether high levels of ROS induced by Menadione treatment is relevant during virus infection. Can the authors discuss this and relate it to virus infection in vivo?

2) MHV68 has a broad tropism, including phagocytic and non-phagocytic cells. Throughout the paper bone marrow derived macrophages were used – are these findings applicable to other non-phagocytic cell types as well?

3) In Figure 4B a western blot for pTBK1 and pIRF3 is shown. The pIRF3 blot doesn't look similar to Figure 2A (pIRF3 seems constitutively expressed). Is this a typo?

4) The authors identify murine STING C147 to be oxidized upon menadione treatment using LC-MS/MS. Additionally, they show that the human STING C148A mutant is unable to form dimers in response to diamide. Even though these 2 lines of evidence indicate that C147 can be oxidized, the graphs presented in Figure 5E do not directly show oxidation of C147. Please include m/z graphs that show evidence for C147 oxidation.

5) The results on MHV68 infection in vivo show relatively modest differences, in particular in conditions treated with NAC. Day 4 post infection in the NAC treated group seem to contain 2 populations of animals, one population that has similar MHV68 levels compared to vehicle group and one population that shows potent viral control. Are there differences within the group? In the Materials and methods section, the authors do not give information on the gender and age used in the experiments. Are there differences in gender and age within the different experimental groups? Does the graph show a representative experiment or are there multiple experiments pooled?

6) Figure 7 shows an insightful model on the proposed function of ROS on STING function. The data presented in the manuscript show that oxidized C147 cannot form dimers (Figure 4E), but the model still suggests a dimer. In addition, the interpretation would be improved if C147 is indicated on the STING molecules. Appropriate modifications are needed.

7) The authors used serum-free media in Figures 4 and 5 but did not elaborate, what is the reason for carrying out those experiments specifically under this condition?

8) It is intriguing to see that the viral loads appear comparable between the WT and STING-KO BMDMs in Figure 4D. Is the same situation with IFN levels?

9) The authors solely relied on Western blotting of STING on native-gel to assess STING oligomerization and activation, other complementary assays such as STING translocation or/and STING/TBK1 complexes should be considered. For example, the STING blot in Figure 5B does not seem correlative of p-TBK1, and menadione alone apparently triggered STING oligomerization, but not p-TBK1.

10) For the in vivo MHV68 infection study, the IFN response should be measured; ideally, STING-KO mice should be used as well.

11) To study the regulatory effect of endogenously produced ROS, the authors treated mouse macrophages with the antioxidant NAC and then infected with MHV68. Under such conditions enhanced interferon-beta induction was detected, whereas MHV68 growth was not significantly affected. Thus, the presented data are not suitable to fully explain the regulatory role of endogenously produced ROS. The authors should consider formulating the respective conclusive chapters more cautiously.

---

## [Author Response]

Essential revisions:1) As mentioned by the authors, herpesviruses have been described to induce ROS in phagocytic cells. In multiple figures the authors show that Menadione increases virus replication in multi-step growth curves in BMDM, indicating that virus-induced ROS does not reach maximum levels. Unclear is, whether high levels of ROS induced by Menadione treatment is relevant during virus infection. Can the authors discuss this and relate it to virus infection in vivo?

The reviewer is correct in that herpesvirus infection does not induce maximum amount of ROS. Unlike some bacterial pathogens, herpesviruses do not potently induce ROS production (Ma et al., 2012). Too high of a ROS level triggers reactivation of γ herpesvirus, which may disrupt the establishment of life-long latency in the host (Ye et al., 2011). As such, it is plausible that viruses evolve to not elicit a maximal amount of ROS production during infection. Importantly, elevated ROS levels in the context of viral infection are highly relevant, as ROS are increased in elderly people as well as in many diseases, including cardiovascular diseases, diabetes mellitus and cancer (Ma et al., 2012). This could suggest that, in these individuals, the antiviral responses are repressed, and γ herpesvirus infection are controlled less efficiently, leading to poorer clinical outcomes (Delgado-Roche and Mesta, 2020).

As suggested by the reviewer, we have added the possible impact of diseases associated with ROS on anti-viral response to the first paragraph of the Discussion section.

2) MHV68 has a broad tropism, including phagocytic and non-phagocytic cells. Throughout the paper bone marrow derived macrophages were used – are these findings applicable to other non-phagocytic cell types as well?

To test whether our findings are applicable to non-phagocytic cells, we performed experiments in mouse embryonic fibroblasts (MEFs). Briefly, we pretreated MEFs with H_2_O_2_ to increase intracellular ROS and induced STING activation with DMXAA. Pretreating MEFs with H_2_O_2_ at the same dose as we treated macrophages (200 µM) did not inhibit *Ifnb* expression but rather slightly increased it. While high levels of ROS inhibit interferon production in macrophages, a similar ROS dose could prime anti-viral responses by causing mitochondrial and genomic DNA fragmentation in non-phagocytic cells (West et al., 2015). This is consistent with the notion that macrophages are more resistant to ROS damages than other cell types because they constantly produce ROS (Virag et al., 2019). This new data has been added to the revised manuscript as Figure 2—figure supplement 3 (subsection “ROS inhibits interferon response upon STING activation”).

3) In Figure 4B a western blot for pTBK1 and pIRF3 is shown. The pIRF3 blot doesn't look similar to Figure 2A (pIRF3 seems constitutively expressed). Is this a typo?

We apologize for the oversight. It was a typo. Rather than pIRF3, this should be labeled as a loading control SDHA. We have corrected the mistake in Figure 4B.

4) The authors identify murine STING C147 to be oxidized upon menadione treatment using LC-MS/MS. Additionally, they show that the human STING C148A mutant is unable to form dimers in response to diamide. Even though these 2 lines of evidence indicate that C147 can be oxidized, the graphs presented in Figure 5E do not directly show oxidation of C147. Please include m/z graphs that show evidence for C147 oxidation.

We apologize for the missing information, and we have included the m/z graphs as Figure 5—figure supplement 1.

5) The results on MHV68 infection in vivo show relatively modest differences, in particular in conditions treated with NAC. Day 4 post infection in the NAC treated group seem to contain 2 populations of animals, one population that has similar MHV68 levels compared to vehicle group and one population that shows potent viral control. Are there differences within the group? In the Materials and methods section, the authors do not give information on the gender and age used in the experiments. Are there differences in gender and age within the different experimental groups? Does the graph show a representative experiment or are there multiple experiments pooled?

In all animal experiments, mice were 8-12 weeks old and randomly assigned to each group in a sex-matched manner prior to the experiment. The data shown in Figure 6B, in which we measured MHV68 infection with menadione treatment, were the results obtained from a pool of 2 independent animal experiments. However, the data initially shown in Figure 6C, in which we measured MHV68 infection with NAC treatment, were from 1 experiment. To better assess the treatment effect of NAC, we conducted an additional animal experiment and updated Figure 6C. In this pooled experiment, where animals were assigned with similar age and gender-matched in each group, NAC treatment still inhibited MHV68 replication. The modest effect of in vivo experiments is consistent with previous reports that PRRs other than cGAS also participate in the control of MHV68 infection (Bussey et al., 2019; Sun et al., 2015). We have updated and provided more details in both the main figure legend, the Materials and methods section and the fifth paragraph of the Discussion section accordingly.

6) Figure 7 shows an insightful model on the proposed function of ROS on STING function. The data presented in the manuscript show that oxidized C147 cannot form dimers (Figure 4E), but the model still suggests a dimer. In addition, the interpretation would be improved if C147 is indicated on the STING molecules. Appropriate modifications are needed.

We thank the reviewer for the opportunity to clarify our findings and model. In our central model, STING exists as a dimer in apo state, but cannot form polymers upon stimulation in the presence of elevated levels of ROS. At resting state, STING is a dimeric protein which is formed by non-covalent bonds and can be disrupted by SDS treatment. Ligand binding alters the conformation of STING and triggers its oligomerization through the formation of covalent disulfide bonds. The disulfide bonds are resistant to SDS treatment but sensitive to reducing reagents. As such, STING proteins that are manifested as monomers and dimers on non-reducing SDS PAGE gels are actually dimers and oligomers, respectively (Ergun et al., 2019). To clarify our findings and model, we added text (subsection “ROS regulate interferon response by inhibiting STING polymerization”) and schematic diagrams to demonstrate theoretical band sizes under various gel conditions (Figure 4—figure supplement 1) and labeled C147 on STING molecules in an updated Figure 7 in the revised manuscript. We also added a statement in the Discussion (fourth paragraph) to indicate that our data does not tell us whether oxidation of STING prevents cGAMP binding and/or conformational change of STING dimers.

7) The authors used serum-free media in Figures 4 and 5 but did not elaborate, what is the reason for carrying out those experiments specifically under this condition?

Hydrogen peroxide (H_2_O_2_) can be decomposed by catalase in serum. In order to maintain a stable concentration of H_2_O_2_, we treated cells with H_2_O_2_-containing serum-free medium for a short period of time. The only exception is when we used H_2_O_2_ for virus growth assay, in which case serum is essential for maintaining the appropriate cell status for virus growth. We have added a specific paragraph regarding chemical treatment to the Materials and methods section in the revised manuscript.

8) It is intriguing to see that the viral loads appear comparable between the WT and STING-KO BMDMs in Figure 4D. Is the same situation with IFN levels?

The comparison between WT and STING-KO viral loads revealed that MHV68 replicated to a significantly higher titer in STING-KO BMDMs (Figure 4D). Notably, menadione treatment increased MHV68 growth in WT macrophages to a similar level as observed in STING-KO macrophages. However, menadione treatment did not further increased viral load in STING-KO macrophages, suggesting that the effects of menadione treatment on virus replication depends on a functional STING. This is consistent with previous report that STING-KO cells do not make significant amounts of interferon after DNA virus infection (Ma et al., 2018).

9) The authors solely relied on Western blotting of STING on native-gel to assess STING oligomerization and activation, other complementary assays such as STING translocation or/and STING/TBK1 complexes should be considered. For example, the STING blot in Figure 5B does not seem correlative of p-TBK1, and menadione alone apparently triggered STING oligomerization, but not p-TBK1.

We agree with reviewer’s comments that additional evidence other than non-reducing SDS-PAGE would be useful. Therefore, we attempted to perform STING translocation experiment, but we were not able to find a satisfying antibody specific for endogenous STING. To overcome this issue, we coimmunoprecipitated TBK1 with STING-specific antibody to analyze the functional state of TBK1. Consistent with our central model, menadione treatment inhibited TBK1 recruitment upon DMXAA stimulation, likely as a result of reduced functional oligomerization of STING. We have included this data as Figure 4—figure supplement 2 in the revised manuscript (subsection “ROS regulate interferon response by inhibiting STING polymerization").

Although STING oligomerization is necessary, it is not sufficient for phosphorylation of TBK1. Consistent with this notion, diamide, an oxidant that triggers STING oligomerization, did not promote TBK1 phosphorylation (Figure 5B). This is likely because apo STING oligomers were not functional, as a conformational change upon a ligand binding is essential for STING activation (Shang et al., 2019). To improve the clarity of the manuscript, we have added relevant explanations in the Results section in the revised manuscript (subsection “ROS oxidization of STING at Cysteine-148 blocks STING activation”).

10) For the in vivo MHV68 infection study, the IFN response should be measured; ideally, STING-KO mice should be used as well.

We appreciate reviewer’s insightful suggestion. To test IFN responses mounted against gammaherpesvirus in vivo, we measured the transcript levels of *Ifnb1* and *Cxcl10* in both WT and Sting-KO mice during MHV68 infection. While *Ifnb1* and *Cxcl10* transcription was significantly upregulated in WT mice upon virus infection, such responses were diminished in mice lacking STING. Consistent with our in vitro data, menadione treatment dramatically inhibited interferon and ISG production mounted during MHV 68 infection. We have included these new data as Figure 6D-F in the revised manuscript (subsection “ROS regulate MHV68 replication in vivo”).

11) To study the regulatory effect of endogenously produced ROS, the authors treated mouse macrophages with the antioxidant NAC and then infected with MHV68. Under such conditions enhanced interferon-beta induction was detected, whereas MHV68 growth was not significantly affected. Thus, the presented data are not suitable to fully explain the regulatory role of endogenously produced ROS. The authors should consider formulating the respective conclusive chapters more cautiously.

The level of endogenous ROS in resting cells is likely low. We observed that neutralization of this low-level endogenous ROS with NAC in vitro was sufficient to increase interferon levels after MHV68 infection or DMXAA treatment. However, this small increase in interferon was not sufficient to change virus replication. We did observe a modest effect of NAC treatment in enhancing MHV68 replication in vivo, likely because of more dynamic cellular respiration in the in vivo system and the prolonged treatment with NAC before and after infection. Even though there is not a large difference in virus replication, we speculate that changes in redox modification of STING could predispose to autoimmunity, particularly in certain genetic backgrounds or perhaps with repeated infections over time leading to chronic increased interferon. A recent study reported GPX4, which is an antioxidant against ROS, facilitates STING activation and is crucial for anti-DNA viral innate response. This is not only consistent with our finding but also emphasizes the important role of ROS in regulating STING function (Jia et al., 2020). Thus, oxidative regulation of STING is important for innate responses against DNA viruses. We have added this information to the sixth paragraph of the Discussion and reworded the eighth paragraph of the Discussion.

References:

Delgado-Roche L, Mesta F. 2020. Oxidative Stress as Key Player in Severe Acute Respiratory Syndrome Coronavirus (SARS-CoV) infection. Arch Med Res 51:384–387. doi:10.1016/j.arcmed.2020.04.019

Ma Z, Ni G, Damania B. 2018. Innate Sensing of DNA Virus Genomes. Annual review of virology 5:341–362. doi:10.1146/annurev-virology-092917-043244